

# A global map of emission clumps for future monitoring of fossil fuel CO₂ emissions from space

Yilong Wang[1,*], Philippe Ciais[1], Grégoire Broquet[1], François-Marie Bréon[1], Tomohiro Oda[2,3], Franck Lespinas[1], Yasjka Meijer[4], Armin Loescher[4], Greet Janssens-Maenhout[5], Bo Zheng[1], Haoran Xu[6], Shu Tao[6], Diego Santaren[1], Yongxian Su[7]

[1]Laboratoire des Sciences du Climat et de l'Environnement, CEA-CNRS-UVSQ- Université Paris Saclay, 91191, Gif-sur-Yvette CEDEX, France
[2]Global Modeling and Assimilation Office, NASA Goddard Space Flight Center, Greenbelt, MD, USA
[3]Goddard Earth Sciences Technology and Research, Universities Space Research Association, Columbia, MD, USA
[4]European Space Agency (ESA), Noordwijk, Netherlands
[5] European Commission, Joint Research Centre, Directorate Sustainable Resources, via E. Fermi 2749 (T.P. 123), I- 21027
Ispra, Italy
[6]Laboratory for Earth Surface Processes, College of Urban and Environmental Sciences, Peking University, Beijing, China
[7]Key Lab of Guangdong for Utilization of Remote Sensing and Geographical Information System, Guangdong Open Laboratory of Geospatial Information Technology and Application, Guangzhou Institute of Geography, Guangzhou 510070, China

*Correspondence to: Yilong Wang (yilong.wang@lsce.ipsl.fr)

**Abstract.** A large fraction of fossil fuel CO₂ emissions occur within "hotspots", such as cities and power plants, which cover a very small fraction of the land surface. Although some of these emission hotspots are monitored closely, there is no detailed emission inventory for most of them. Space-borne imagery of atmospheric CO₂ has the potential to provide

independent estimates of CO₂ emissions from hotspots. But first, what is a hotspot needs to be defined for the purpose of satellite observations. The proposed space-borne imagers with global coverage planned for the coming decade have a pixel size on the order of a few square kilometers, and a XCO₂ accuracy and precision of <1 ppm for individual pixels. This resolution and precision is insufficient to provide a cartography of emissions for each individual pixel. Rather, the integrated emission of the diffuse emitting area and the intense point sources are sought. In this study, we address the question of the

global characterization of area and point fossil fuel CO₂ emitting sources (those hotspots are called emission clumps hereafter) that may cause coherent XCO₂ plumes in space-borne CO₂ images. An algorithm is proposed to identify emission clumps worldwide, based on the ODIAC global high resolution 1 km fossil fuel emission data product. The clump algorithm selects the major urban areas from a GIS (geographic information system) file and two emission thresholds. The selected urban areas and a high emission threshold are used to identify clump cores such as inner city areas or large power plants. A

low threshold and a random walker (RW) scheme are then used to aggregate all grid cells contiguous to cores in order to define a single clump. With our definition of the thresholds, which are appropriate for a space imagery with 0.5 ppm precision for a single XCO₂ measurement, a total of 11,314 individual clumps, with 5,088 area clumps and 6,226 point-source clumps (power plants), are identified. These clumps contribute 72% of the global fossil fuel CO₂ emissions according to the ODIAS inventory. The emission clumps is a new tool for comparing fossil fuel CO₂ emissions from



different inventories, and objectively identifying emitting areas that have a potential to be detected by future global satellite imagery of $XCO_2$. The emission clump data product is distributed from https://doi.org/10.6084/m9.figshare.7217726.v1.

## 1 Introduction

Monitoring the effectiveness of emission reductions after the Paris Agreement on Climate (UNFCCC, 2015) requires frequently updated estimates of fossil fuel $CO_2$ emissions and a global synthesis of these estimates. The need for emission monitoring goes beyond national estimates, as many cities and regions have set concrete objectives to reduce their greenhouse gas emissions. The $CO_2$ emissions (direct and indirect) related to final energy use in cities are estimated to be 71% of the global total (IEA, 2008; Seto et al., 2014). In addition, power plants account for ~40% of direct energy-related $CO_2$ emissions (Tong et al., 2018), and are subject to regulations that require a regular reporting of their emissions. The contribution from cities and power plants to national and global mitigation efforts is thus critical (Creutzig et al., 2015; Shan et al., 2018).

Research to quantify emissions based on prior information on their magnitude and distribution, atmospheric $CO_2$ concentration measurements and atmospheric transport models is a branch of science called atmospheric $CO_2$ inversions. Inversions of fossil fuel $CO_2$ emissions have used in-situ surface networks around cities (Bréon et al., 2015; Lauvaux et al., 2016; Staufer et al., 2016), but the deployment of a network around each city may be impractical. Alternatively, it is possible to measure vertically integrated columns of dry air mole fractions of $CO_2$ ($XCO_2$) from satellites passing over emission hotspots. Satellite measurements offer the advantage of global spatial coverage, but research studies consistently outlined that satellite $XCO_2$ measurements need to have a high precision (< 1 ppm) and a spatial sampling at high resolution (< 2-3 km horizontal resolution) (Bovensmann et al., 2010; O'Brien et al., 2016). For example, the Greenhouse Gases Observing Satellite (GOSat-2) aims to measure $XCO_2$ at 0.5 ppm precision (https://directory.eoportal.org/web/eoportal/satellite-missions/g/gosat-2). The single sounding random error in $XCO_2$ from the Orbiting Carbon Observatory 2 (OCO-2) is on the order of magnitude of 0.5 ppm (Eldering et al., 2017; Chatterjee et al., 2017). $XCO_2$ measurements from selected 10 km wide OCO-2 tracks downwind of large power plants were used to quantify their emissions by fitting observed $XCO_2$ plumes with Gaussian dispersion models (Nassar et al., 2017). According to Nassar et al., (2017), the uncertainties in the emissions from three selected U.S. power plants were constrained within 1–17% of reported daily emission values. The primary scientific goal of the OCO-2 mission was to estimate natural land and ocean carbon fluxes, and tracks overpassing power plants are very sporadic, given the narrow swath width and frequent clouds. In order to improve the sampling of the atmosphere, $XCO_2$ imagers (e.g. passive spectral-imagers in the short wave infrared spectrum) are under study. The list includes the Geostationary Carbon Observatory (GeoCARB) mission (Polonsky et al., 2014), the OCO-3 instrument on board the International Space Station capable of pointing to chosen emitting areas (https://www.nasa.gov/mission_pages/station/research/experiments/2047.html) and a constellation of low earth orbiting





(LEO) imagers with a swath of a few hundred kilometers planned as future operational missions within the European Copernicus Program (Ciais et al., 2015).

The ability of imaging instruments to reduce uncertainty on $CO_2$ emissions was investigated by atmospheric inversions with pseudo-data, that is, Observing System Simulation Experiments (OSSEs), but only for case studies of limited duration. OSSEs were performed for large cities (Broquet et al., 2018; Pillai et al., 2016), single power plants (Bovensmann et al., 2010) or for a region encompassing several cities (O'Brien et al., 2016). An OSSE study with one LEO imager over Paris (Broquet et al., 2018) solved for emissions during the 6 h before a given satellite overpass. Their results showed that the uncertainty (~25%) in the 6 h mean emissions in the prior estimates could be reduced to less than 10% during few days when the wind speed is low and there is not much cloud. The results of such case studies are informative about the potential of satellite observations in quantifying fossil fuel $CO_2$ emissions, but do not inform systematically about how many hotspots, and which fraction of emissions worldwide could be constrained with $XCO_2$ imagers.

A prerequisite for assessing the capability of satellite imagers is to have a high resolution global map of fossil fuel $CO_2$ emissions. We use in this study the ODIAC map at 30×30 arc-seconds (~ 1 km×1 km) (Sect. 2.1). Not all the emitting 1 x 1 km land grid-cells of such a map will have emissions sufficiently intense to produce a $XCO_2$ plume detected with a satellite (Nassar et al., 2017; Hakkarainen et al., 2016). On the other hand, a cluster of contiguous emitting grid cells will create a stronger plume than a single emitting grid cell, so that the uncertainty on the sum of emissions from a cluster could be reduced with space-borne measurements. This poses the research question of how to define those clusters of emitting pixels (called emission clumps hereafter) who will generate individual $XCO_2$ plumes being detectable from space. The emission clumps should include intense area sources and large isolated point sources (e.g. power plants, large factories). Using political and administrative area of cities to define clumps does not work for this purpose because the same administrative area may contain separate large point sources or multiple hotspots forming separable plumes, as well as areas with no or little emission. The definitions of emitting areas differ among inversion studies. Broquet et al. (2018) estimated emissions from the Île de France region, while Pillai et al. (2016) defined their emitting region as an area of 100 km×100 km around Berlin. The arbitrary choice of emitting areas across studies make the comparison of their results difficult and are not applicable worldwide. This justifies the need for a systematic and objective definition of emission clumps that constitute observing targets for satellites.

The algorithm for calculating emission clumps developed in this study is inspired by research on mapping urban area and socio-demographic activities (Li and Zhou, 2017; Elvidge et al., 1997; Zhou et al., 2015; Su et al., 2015; Doll and Pachauri et al., 2010; Letu et al., 2010). The corresponding algorithms can be grouped in classification-based or threshold-based. Classification-based algorithms use dataset such as normalized difference vegetation index (NDVI) and normalized difference water index (NDWI) to train a machine-learning model to classify urban and non-urban areas (Cao et al., 2009; Huang et al., 2016). Threshold-based algorithms classify urban grid cells where some continuous variables (e.g. nighttime lights) are above a given threshold (Elvidge et al., 1997; Liu and Leung, 2015; Li et al., 2015; Liu et al., 2015). In



threshold-based methods, given the high spatial heterogeneity of urbanization and urban forms, efforts have been devoted to
105   find local optimal thresholds, such as the "light-picking" approach to find a local nighttime background light surrounding a
target grid cell (Elvidge et al., 1997), or determining local thresholds by matching local/site-based surveys and
land-use/land-cover (LULC) datasets (Zhou et al., 2014).

The problem of characterizing $CO_2$ emission clumps posed here consists in delineating all areas that have a potential to
generate detectable atmospheric $XCO_2$ plumes. "Detectable" means here that the concentration within a plume formed by a
110   clump should be large enough compared to the surrounding background in $XCO_2$ images of typical spatial resolution of ≈1
km. The magnitude of a minimum detectable $XCO_2$ enhancement in a plume (relative to the surrounding background)
depend on individual $XCO_2$ sounding precision. Such sounding precision should be of similar order of magnitude worldwide,
although the solar zenith angle, aerosol loads, surface albedo etc. will affect it (Buchwitz, et al., 2013). In this context,
contrary to the algorithms used for mapping urban area, common global minimum emission thresholds for land grid cells
115   forming a clump are relevant.

This study aims to provide a global dataset of fossil fuel $CO_2$ emission clumps for high-resolution atmospheric
inversions that will use $XCO_2$ imager data. Such a dataset can be used for OSSE studies to compare different imagery
observation concepts for constraining fossil fuel $CO_2$ emissions at clump scale over the whole globe. We propose an
approach that combines a threshold-based and an image-processing algorithm. Section 2 describes the high-spatial resolution
120   global emission map upon which clumps are calculated, and the algorithm to delineate the clumps worldwide. The spatial
distribution and extent of the resulting clumps throughout the globe are described in Sect. 3 and are compared with clumps
diagnosed by applying the same algorithm to other emission maps. Section 4 discusses the sensitivity of the resulting clumps
to the precision of $XCO_2$ measurements and future applications of this global dataset. Section 5 describes the data availability.
Conclusions are drawn in Sect. 6.

## 2. Methodology

### 2.1 ODIAC fossil fuel $CO_2$ emission map

We use the high-spatial resolution (30" × 30" ≈ 1 km × 1 km) global annual fossil fuel $CO_2$ emission map for the year
2016 from the Open Source Data Inventory of Anthropogenic $CO_2$ Emission (ODIAC, version 2017) (Oda and Maksyutov,
2011; Oda et al., 2018) for calculating clumps. To our knowledge, it is the only emission map with global coverage and a
spatial resolution high enough to match the pixel size of ≈ 1 km of atmospheric $XCO_2$ imagers. We chose the year 2016
assuming that the emission spatial distributions do not change significantly from year to year. The ODIAC dataset provides
emissions from power plants based on the CARMA database (Carbon Monitoring and Action, http://carma.org). Emissions
from these point sources were spatially allocated to the exact locations from CARMA. Emissions from other sources
(industrial, residential, commercial sectors and daily land transportation) were estimated by subtracting the sum of emissions



from power plants in each country from the national totals given by the Carbon Dioxide Information and Analysis Center
      (CDIAC) (Boden et al., 2016). Annual emissions in each country excluding power plants were spatially distributed at 30"
      spatial resolution using nighttime light fields from the Defense Meteorological Satellite Program (DMSP) satellites. ODIAC
      has been used in atmospheric inversions to monitor $CO_2$ emissions from cities (Oda et al., 2018; Lauvaux et al., 2016).

      Because $CO_2$ produced by emissions is quickly dispersed by transport, $XCO_2$ plumes sampled at a given time by a
satellite image usually relate to emissions that occurred few hours before its acquisition (Broquet et al., 2018). Here we
      assumed an equator crossing time around 11:30 local time for LEO imagers on Sentinel missions (Buchwitz et al., 2013;
      Broquet et al., 2018), so that $XCO_2$ plumes sampled by these imagers are from morning emissions. Different overpass times
      are also possible for other satellites. For example, Equator crossing times of OCO-2 and GOSAT are 13:00-13:30 local time.
      Geostationary imagers may provide a better temporal coverage of the emissions; e.g. GeoCARB images are considered to
sample a city for multiple times within a day (O'Brien et al., 2016).

      We adopt here a focus on planned Sentinel LEO imagers, and thus use annual average of morning emissions for
      calculating emission clumps. To estimate morning emissions, we combined the ODIAC emission maps with the hourly
      profiles from the Temporal Improvements for Modeling Emissions by Scaling (TIMES) product (Nassar et al., 2013). In
      TIMES, the hourly profiles were provided as 24 scaling factors for each hour of the day that can be multiplied by daily
average emissions to derive hourly emissions. Hourly scaling factors of TIMES were derived for residential, commercial,
      industrial, electricity production and mobile on-road sectors from the bottom-up model of fossil fuel $CO_2$ emissions Vulcan
      v2.0 over the US (Gurney et al., 2009) with mobile non-road, cement manufacture and aircraft assumed temporally constant.
      The TIMES dataset also gives hourly scaling factors for other 19 high-emitting countries. These profiles were weighted by
      the emissions fraction in each sector from EDGAR to determine hourly profiles of total $CO_2$ emissions. The US and other 19
high-emitting countries are called proxy countries. Other countries in the world were assigned the same profiles than one of
      the proxy countries, accounting for standard international time zones, local socio-demographic patterns (e.g. time of day
      when people start to work, weekend defined according to different religions). The TIMES hourly profiles were derived at
      national scale (assuming identical hourly profiles within a country) and then shifted by hourly offsets according to local solar
      time to approximate the variability related to geophysical cycles. The original TIMES hourly profiles at $0.25°×0.25°$
resolution were downscaled at the spatial resolution of ODIAC, assuming the same profiles within each $0.25°×0.25°$ grid cell.
      For calculating clumps based on morning emissions, we multiplied the annual mean emission rate (unit: g C m$^{-2}$ hr$^{-1}$) in each
      grid cell of ODIAC by the average scaling factors of emissions between 6:00-12:00 local time.

### 2.2 Calculation of emission clumps

      The emission clumps from point sources and intense area sources in ODIAC are separated in this study. In ODIAC, the
point sources only refer to power plants in the CARMA database. Before clumps are calculated, Fig. 1 illustrates the ranked
      distribution of emission rates during morning hours from point sources (red) and other grid cells (blue). Excluding emissions



from point sources, the maximum emission rate of emitting grid cells from area sources is 20.7 g C m$^{-2}$ hr$^{-1}$ and most grid cells including point sources have much larger emission rates than this value. In total, 35% of the global total emissions are from 12433 30"×30" grid cells encompassing at least one point source.


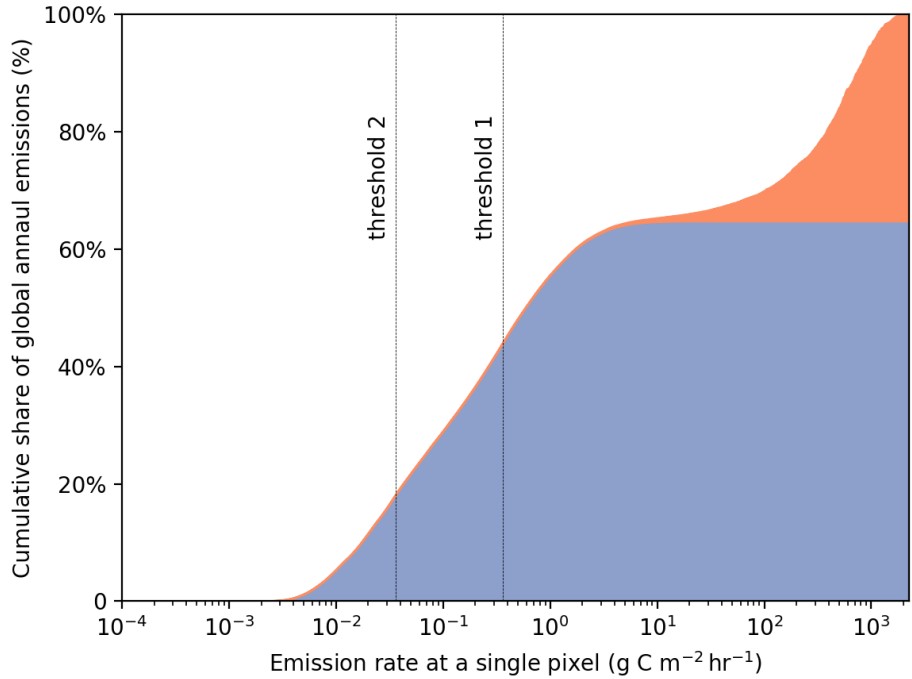

**Figure 1** Cumulative distribution of mean emission rates during morning hours in ODIAC for power plants (red) and area sources (blue). The y-axis represents the cumulative share of global total annual emissions at each level of emission rate for a single land grid cell (x-axis). The vertical dash lines are the two thresholds used in the clump algorithm (see text).


      Figure 2 shows the flowchart of the clump algorithm. Fig. 3 illustrates how it operates for a small domain as an example. Firstly, only grid cells encompassing point sources with an emission rate larger than threshold-1 are considered. This threshold is chosen as 0.36 g C m$^{-2}$ hr$^{-1}$, based on the argument that, even without any atmospheric horizontal transport, emissions lower than this threshold over 6 hours would generate a local XCO$_2$ excess of less than 0.5 ppm, the practical limit

of individual sounding precision from current satellites (see Appendix for the detailed computation). This is illustrated in Fig. 3b by the red grid cell labeled as 1 and 2. There are 6226 grid cells in ODIAC2017 who encompass at least one power plant and whose emission rates are above threshold-1, which account for >99.99% of total emissions of all CARMA power plants globally.

      Secondly, emissions clumps from area sources are calculated. We combine two data streams to calculate area clumps: 1)

the administrative division of major urban areas; and 2) two thresholds (threshold-1 and threshold-2 detailed below) applied to the grid cells of ODIAC. We assume that a group of emitting pixels encompassing some adjacent high emitting pixels



(forming a core of the emission clump) and their surroundings will generate an individual plume in $XCO_2$. The urban area and the high threshold (threshold-1) define the cores of each emission clump, while threshold-2 defines the lower limit of surrounding emitting pixels to be potentially included in the clumps. The four steps to compute area sources emission clumps are detailed as below.

1) The value of threshold-2, above which emissions of a single emitting grid cell is selected to be potentially included in a clump, is chosen as 0.036 g C m$^{-2}$ hr$^{-1}$, a factor of 10 lower than threshold-1. The sum of emissions from grid cells above threshold-2 represents 82% of global total emissions (including point sources). Grid cells below threshold-2 are never included in any emission clump. Grid cells whose emission rates are above threshold-2 are illustrated in Fig. 3a by the yellow and orange grid cells;

2) We then used the urban area GIS (geographic information system) file from the Environmental Systems Research Institute (ESRI, https://www.arcgis.com/home/item.html?id=2853306e11b2467ba0458bf667e1c584) to locate the geographic positions of major urban areas. ESRI contains 3615 separated urban areas, defined independently from the ODIAC emission map. We found 2017 ESRI urban areas containing at least one grid cell with emission above threshold-1. The remaining 1598 ESRI urban areas are not considered hereafter. An illustration of one of the 2017 selected ESRI urban area is shown in Fig. 3c by the grid cells labeled as 3. Figure 4a-4c (solid lines) shows three examples of ESRI urban areas for major cities in Europe, North America and China. The grid cells within the ESRI urban area whose emission rates are above threshold-1 define the cores of the clumps.

3) Although the ESRI GIS file cover large cities of the world, smaller populated areas, like towns on the southeast coast of China that may also generate detectable plumes, are missed by ESRI map. This calls for a complementary step to identify non-ESRI emitting clumps. For the calculation of those non-ESRI clumps, we apply threshold-1 of 0.36 g C m$^{-2}$ hr$^{-1}$ to all grid cells that are not selected in the previous step as part of any ESRI core. Contiguous non-ESRI grid cells above threshold-1 form non-ESRI core of clumps. These non-ESRI core grid cells must be spatially distinct from the ESRI core grid cells. If they are adjacent to any ESRI core, they are absorbed by the ESRI ones. A total of 3071 non-ESRI cores are calculated, as shown in Fig. 3d by the grid cells labeled as 4;

4) After ESRI and non-ESRI clump cores are defined, we aggregate all the emitting grid cells whose emission rates are larger than threshold-2 in their vicinity to form a clump. An ensemble of grid cells with emissions higher than threshold-2 in a domain with $N$ cores are attributed to $N$ distinct emission clumps. The attribution of a grid cell to a given core is calculated based on the spatial gradients of emissions and the distance between the emitting grid cells by using a "random walker" (RW) algorithm (Grady, 2006). RW is a type of algorithm used in the field of image segmentation, i.e. recognizing different segments/objects in a picture or photograph. This step is illustrated in Fig. 3e by the grid cells in light yellow.

The RW algorithm defines the probability of each grid cell to belong to some known labeled "seeds" (i.e. the cores defined in steps 2 and 3 in this study). This algorithm imagines that a random walker start from each grid cell to be labeled (in this study, the grid cells whose emissions that are above threshold-2 but not included in the cores). The probability that





the walker will arrive at each known seeds, following the easiest path, are computed. The undefined grid cells are assigned to the seed that has the highest probability to be reached by the walker. Specifically, in this study, we define the probability that the walker move between two neighboring grid cells using an exponential decaying function of the $\ell^2$ norm of the log-transformed local gradients in emissions (Grady, 2006):

$$w_{ij} = e^{-\beta(g_i - g_j)^2} \tag{1}$$

where $w_{ij}$ is the probability of motion between neighboring grid cells $i$ and $j$, $g_i$ and $g_j$ are image intensity (defined as the log-transformed emission rate in this study), and $\beta$ is a free penalization parameter for the motion of random walker (the greater the $\beta$, the more difficult the motion). In this study, $\beta$ only impacts how the undefined grid cells are assigned to the cores. It balances the effect of local gradients and the distance of the path from the undefined grid cells to the seeds: the larger the gradients along a path between the undefined grid cells and the seeds, the smaller probability that the walker would
move; and the longer the path, the smaller the probability that the walker would arrive at corresponding seeds. Larger $\beta$ will lead to larger impact of emission gradients than that of distance. In this study, $\beta = 13\ \sigma_g^{-1}$, where $\sigma_g$ is the standard deviation of the emission rates at all the grid cells in ODIAC. In general, the algorithm can effectively separate different clusters of grid cells with different spatial distributions. For instance, a clump with a flat distribution of emissions and a clump (of similar size as the former one) with more skewed emissions are separated near the steepest gradients. This assumes that large
emission gradients will generate large gradients in $XCO_2$ (given similar meteorological condition for neighboring clumps), and that different $XCO_2$ plumes are separable where the $XCO_2$ gradients are the largest.

After the RW algorithm, grid cells above threshold-2 that are not contiguous to any core are discarded. This removes 10% of the total from the 82% of global emissions defined in step 1. As a result, 72% of the global emissions are included in the emission clumps (see more detailed discussion below).

All the computation are made under the Python version 2.7 environment (Python Software Foundation, http://www.python.org) and the RW algorithm is from package "scikit-image" version 0.14dev (http://scikit-image.org/).





**Figure 2** The flow chart of emission clumps calculation. The colors qualitatively illustrate grid cell emission rates from low
(light green) to high (red)





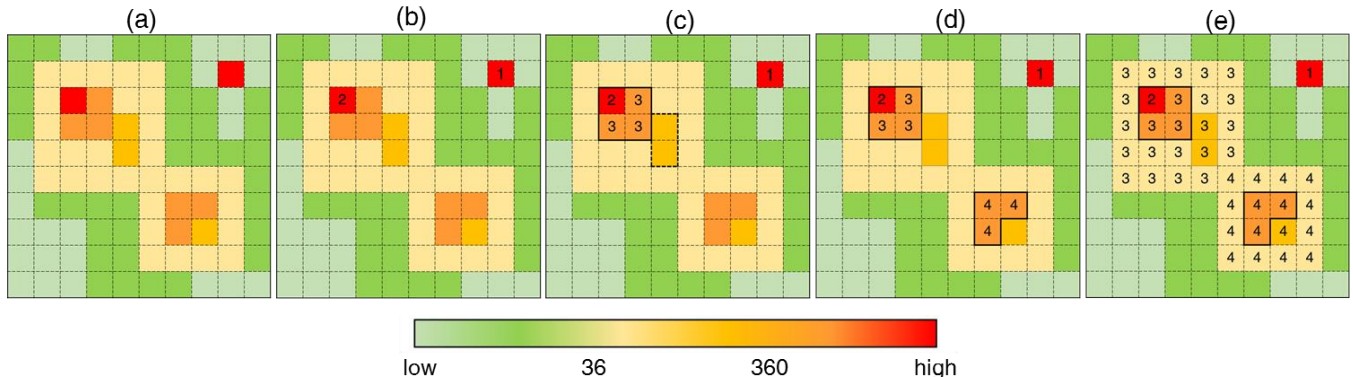

**Figure 3** The processes of defining emission clumps. The colors qualitatively illustrate the emission rates from low (light green) to high (red). a) the emission field; b) 2 power plant (red grid cells) is defined as two individual clumps, labelled as 1 and 2; c) The ESRI urban area is outlined by bold solid and dashed lines, but the ESRI core is labelled as 3 only for grid cells whose emission rates are above threshold-1; d) the orange area represent grid cells whose emissions are above threshold-1 to form a non-ESRI core, labelled as 4; e) each light-yellow grid cell is assigned to one of the clump cores using the RW algorithm (see the main text). Note that one power plant (labelled 2) is located within the ESRI urban area, but is identified as a different emission clump from the ESRI clump (labeled as 3 in Fig. 3e)

## 3. Results

### 3.1 Emission clumps defined on ODIAC emission map

Figure 4 shows three regional clumps near Paris (France), New York (USA) and Beijing (China). The clumps near Paris are well isolated from each other. There are more emission clumps in the New York region. Because some clumps are close to each other in this region (e.g. New York and Clifton), their plumes will only be distinct when the wind direction is roughly perpendicular to the direction of the line connecting clumps (i.e. from southwest to northeast or the opposite for New York and Clifton). Near Beijing, there are a larger number of clumps than in the other two regions and their distribution is also more complex.

Table 1 summarizes the clumps calculated for the globe, Europe (European Russia included), China, North America, South America, Africa, Australia and Asia (China excluded). In total, our algorithm calculates 11314 clumps, including 6226 point sources, 2017 ESRI clumps, and 3071 non-ESRI clumps. The clump with largest emission is Shanghai, which emits 47 Mt C per year. A large fraction of the non-ESRI clumps is found within China mainly located near the southeastern coast, which may be explained by the recent rapid urbanization (Shan et al., 2018; Wang et al., 2016) in this region. This is not documented by the ESRI map. The large number of non-ESRI clumps in China highlights the necessity to consider emitters outside the major cities (at least) in this country. In addition, the mean area of an emission clump is larger in China than over other continents/regions. This is because the southeast coast of China is densely populated even at rural places (yellow-green



in Fig. 4e), and because the emission rates per capita is also high in China (Janssens-Maenhout et al., 2017). As a result, our algorithm finds: 1) more cores (of non-ESRI clumps) in China than other regions; and 2) larger area with emission rates larger than threshold-2.

Figure 5 shows the locations and annual emissions of the clumps. The densities of emission clumps are high in Europe, the East Coast of US, the East Coast of China and India. Fig. 6 shows the fractions of total emissions allocated to different clump categories. Globally, 27% of the clumps are calculated as non-ESRI, but the total emission from these clumps is less than 13% of the total emissions. Point sources form 55% of the total number of clumps and 44% of the total emissions. In China, however, point sources contribute only 21% of the total number of clumps and 39% of the total emissions, which may

be explained by the fact that the power plants in China considered in CARMA dataset (and thus in ODIAC) are limited to the few larger power plants. Fig. 7 shows the cumulative distribution of the number of clumps and their emission for a few regions. Among ESRI clumps, 66% of them have an annual emission below 1 Tg C yr$^{-1}$, but the cumulative emission from these low emitting clumps only account for 22% of the total emissions from all ESRI clumps. The inflexion point in Fig. 7 (when the cumulative distribution curve turns from nearly 0% to a fast increase) indicates the importance of clumps whose

annual emissions are above this value. For non-ESRI clumps and point sources, the inflexion points are near 0.1 Tg C yr$^{-1}$.





**Figure 4** Emission clumps near Paris (a and d), Beijing (b and e) and New York (c and f). In a-c, solid lines depict the urban areas from ESRI product. Colored patches depict the clump area resulting from the algorithm defined in this study. In d-f, solid lines depict the boundaries of final clumps (boundary of colored patches in a-c). Colored fields in d-f show the emissions from ODIAC product. Light dashed lines indicate 1 °×1 °grids.



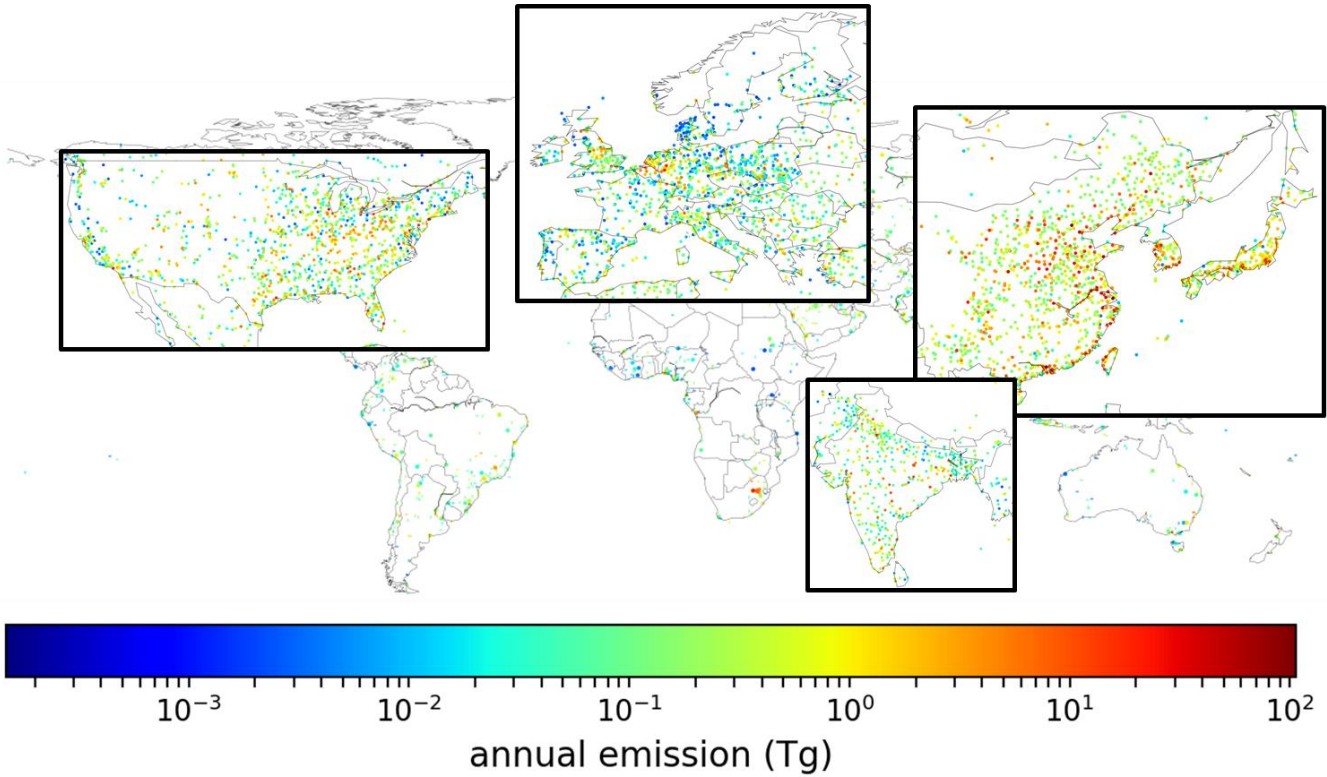

**Figure 5** The spatial distribution of emission clumps all over the globe. The inserted plots zooms over 4 regions that contain most of the clumps.




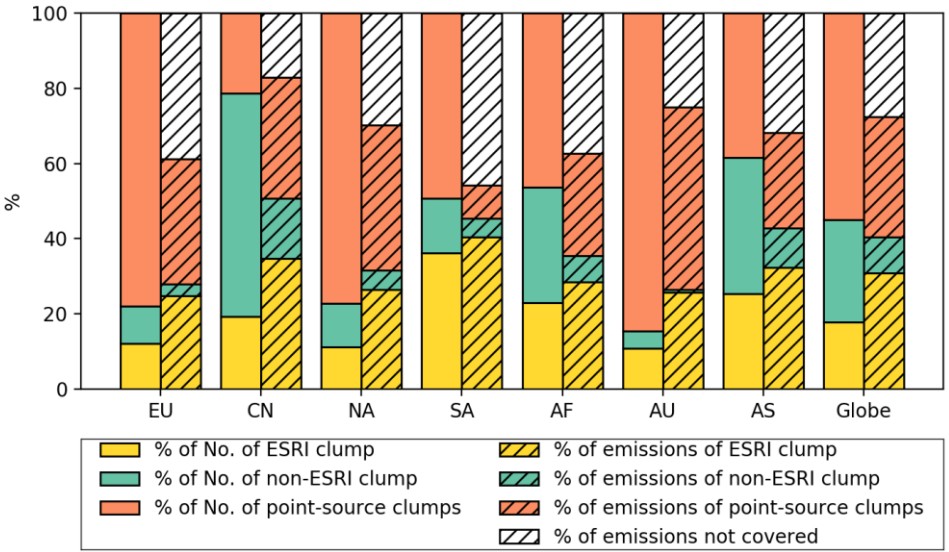

**Figure 6** The fraction of the number (bars) and the fraction of emissions (hatched bars) found in the three types of clumps for European continent (European Russia included), China, North America (NA), South America (SA), Africa, Australia, Asia with China excluded (AS) and over the globe. The three colors represent ESRI clumps (yellow), non-ESRI clumps (green) and point-source clumps (red), respectively. The white-hatched bars indicate the fraction of ODIAC emissions that

are not allocated into any clump by the algorithm.

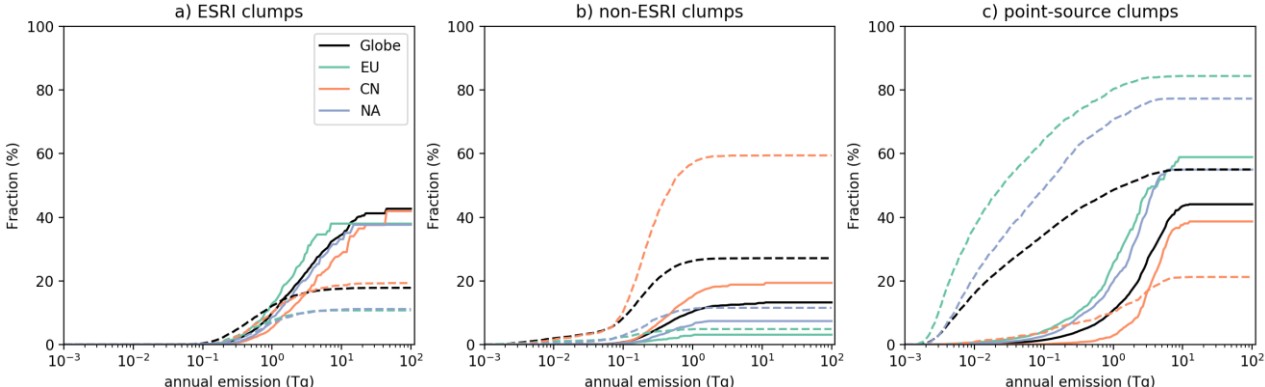

**Figure 7** Cumulative distributions of the number (dashed lines) of emission clumps and of the emissions (solid lines) of the clumps for three categories of clumps (see text).


**Table 1** Characteristics of clumps defined in this study for the globe, European continent (European Russia included), China, North America (NA), South America (SA), Africa, Australia and Asia with China excluded (AS).

| | Globe | Europe | China | NA |
|---|---|---|---|---|



| | | | | |
|---|---|---|---|---|
| Total number of clumps | 11314 | 2470 | 2091 | 2616 |
| Number of ESRI clumps | 2017 | 300 | 404 | 292 |
| Number of non-ESRI clumps | 3071 | 243 | 1243 | 302 |
| Number of point-source clumps | 6226 | 1927 | 444 | 2022 |
| Mean area of one area clump ($km^2$) | 196 | 125 | 337 | 137 |
| Maximum area of one area clump ($km^2$) | 10356 | 6874 | 5762 | 5568 |
| Mean emission budget of one clump ($Tg\ yr^{-1}$) | 0.57 | 0.31 | 1.02 | 0.45 |
| Maximum emission budget of one clump ($Tg\ yr^{-1}$) | 47 | 17 | 47 | 15 |
| Minimum emission budget of one clump ($Tg\ yr^{-1}$) | $9.9\times10^{-4}$ | $9.9\times10^{-3}$ | $21\times10^{-4}$ | $19\times10^{-4}$ |
| Clump that has the largest annual emission | Shanghai | Moscow | Shanghai | Los Angeles |
| Fraction of emissions from defined clumps to total emission | 72% | 60% | 84% | 70% |
| Share of urban $CO_2$ emissions to regional total in IEA report | 67% | 69% | 75% | 80% |
| Share of urban energy use to regional total in GEA report | 76% | 77%[1] | 65%[2] | 86% |

(Table 1 continued)

| | SA | Africa | Australia | AS |
|---|---|---|---|---|
| Total number of clumps | 477 | 470 | 110 | 2784 |
| Number of ESRI-urban clumps | 172 | 108 | 12 | 705 |
| Number of non-ESRI clumps | 69 | 144 | 5 | 1007 |
| Number of point-source clumps | 235 | 218 | 93 | 1072 |
| Mean area of one area clump ($km^2$) | 186 | 183 | 133 | 229 |
| Maximum area of one area clump ($km^2$) | 4303 | 3438 | 3113 | 10356 |
| Mean emission budget of one clump ($Tg\ yr^{-1}$) | 0.35 | 0.43 | 0.69 | 0.63 |
| Maximum emission budget of one clump ($Tg\ yr^{-1}$) | 12 | 11 | 6.8 | 22 |
| Minimum emission budget of one clump ($Tg\ yr^{-1}$) | $20\times10^{-4}$ | $26\times10^{-4}$ | $21\times10^{-4}$ | $17\times10^{-4}$ |
| Clump that has the largest annual emission | Buenos Aires | Johannesburg | Melbourne | Riyadh |



| | | | | |
|---|---|---|---|---|
| Fraction of emissions from defined clumps to total emission | 52% | 62% | 76% | 69% |
| Share of urban $CO_2$ emissions to regional total in IEA report | - | - | 78% | - |
| Share of urban energy use to regional total in GEA report | 85% | 69%[3] | 78% | 63%[4]- |

[1] Arithmetic mean of values for Western Europe and Eastern Europe
[2] In GEA report, this value correspond to China and Central Pacific Asia
[3] Arithmetic mean of values for Sub-Saharan Africa, North Africa and Middle East
[4] Arithmetic mean of values for Pacific Asia and South Asia

### 3.2 Emission clumps based on other emission maps

The clump results obviously depend on the input emission field. The ODIAC map is chosen as a reference because it is the only global map with a spatial resolution of ~ 1 km that we are aware of. But there are other emission products with coarser resolution or having only regional coverage. To test the dependency of calculated clumps on the choice of emission map, we apply the algorithm to three alternative global emission maps and two regional emission maps (Table 2). The three global emission maps are: PKU-$CO_2$ v2 (Wang et al., 2013), FFDAS v2.0 (Rayner et al., 2010; Asefi-Najafabady et al., 2014), EDGAR 4.3.2 (Janssens-Maenhout et al., 2017). The two regional emission maps are: the Multi-resolution Emission Inventory (MEIC) v1.2 for China (http://meicmodel.org/; Zheng et al., 2018) and the VULCAN inventory (Gurney et al., 2009) v2.2 for the contiguous U.S. The resolutions of these emission maps are 0.1° or 10 km (Table 2), that is, about 12 times coarser than ODIAC. Note that some small (in terms of area) groups of grid cells with high emission rates at finer scale than 0.1° are averaged at coarse grid cells in these coarse-resolution maps. The clumps derived from these alternative emission maps thus have a tendency to miss small clumps, compared to ODIAC. However, the comparison of the results for the largest clumps is still indicative of the robustness of the clump definition. The years of the additional emission maps are different from the year of ODIAC (Table 2) because some institutions do not have released their emission maps for 2017. We scale the different emission maps to the same national totals as ODIAC and we assume that the spatial distribution of clumps do not change significantly at continental and global scales so that the differences in the year for different emission maps is not expected to have strong impacts on the clump results. We compare the fractions of emissions in alternative maps (X) covered by the clumps calculated from these map (X-clumps) with the fraction covered by ODIAC-clumps to see whether the ODIAC-clump results miss significant emissions from X. Because the resolution of ODIAC and alternative emission maps are different, when computing the X emissions covered by ODIAC-clumps, we downscale map X to 30", assuming that emissions are distributed uniformly within each 0.1° or 10 km grid cell. Since the actual distribution of emissions within each 0.1° or 10 km grid cell is probably not uniform, this computation tend to overestimate the differences between ODIAC-clumps and X-clumps.



**Table 2** The alternative emission maps used to compare with the results of ODIAC

| Emission product | Coverage | Resolution | Year | Reference |
|---|---|---|---|---|
| EDGAR 4.3.2 | Global | 0.1°×0.1 ° | 2010 | Janssens-Maenhout et al., 2017 |
| PKU-CO$_2$ v2 | Global | 0.1°×0.1 ° | 2010 | Wang et al., 2013 |
| FFDAS v2.0 | Global | 0.1°×0.1 ° | 2009 | http://hpcg.purdue.edu/FFDAS/Map.php; Rayner et al., 2010; Asefi-Najafabady et al., 2014 |
| MEIC v1.2 | Global | 0.1°×0.1 ° | 2010 | http://meicmodel.org; Zheng et al., 2018 |
| VULCAN v2.2 | 74% | 10 km×10 km | 2002 | Gurney et al., 2009 |

Each 30" grid cell is classified into a confusion matrix (CM) with 4 categories: 1) grid cell belongs to ODIAC-clump and X-clump (true positive, TP); 2) grid cell belongs to ODIAC-clump but not to X-clump (false positive, FP); 3) grid cell belongs to X-clump but not to ODIAC-clump (false negative, FN); and 4) grid cell neither in ODIAC-clump nor in X-clump
(true negative, TN). The fractions of emissions in each CM category are computed for different regions. This comparison mainly allows us to verify whether the clumps delineated by the two thresholds are consistent using ODIAC and other maps.

We also checked the consistency of ESRI clumps between ODIAC-clump and X-clumps with a similar CM. Each grid cell is classified into four categories: 1) grid cell belongs to the same ESRI clump in ODIAC and X (ESRI-TP); 2) grid cell belongs to ESRI clumps in both ODIAC and X, but does not belong to the same ESRI clump (ESRI-DIFF); 3) grid cell only
belongs to an ESRI clump either in ODIAC or X (ESRI-FALSE); and 4) grid cell does not belong to any ESRI clump in ODIAC nor in X (ESRI-TN). Consistency for non-ESRI clumps is not really expected because X-clumps tend to miss small clumps because of the underlying coarser-resolution maps. Consistency is not calculated for point-source clumps because not all emission products explicitly provide names for each power plant, making it difficult to determine whether the power plants from different maps within a same grid cell are the same infrastructure.

VULCAN is arguably the best emission map for the US, for the use of the large amount of accurate data from local to national scales. PKU-CO$_2$-v2 and MEIC v1.2, derived by Chinese institutions, used the exact locations of power plants and factories in China and detailed information of fuel consumption of each power plants and factories to estimate the point sources. They also used provincial data to distribute the non-point source emissions, resulting in more accurate estimates in the distribution of Chinese emissions than other global maps (Wang et al., 2013). EDGAR v4.3.2, developed by the Joint
Research Center under the European Commission's service, has more realistic emission estimates in Europe. Therefore, we focus the clump consistency analysis between ODIAC and EDGAR v4.3.2 for Europe, between ODIAC, PKU-CO$_2$-v2 and MEIC v1.2 for China, and between ODIAC and VULCAN v2.2 for the US.



Figure 8 shows the results of the CM analysis. In general, there is a considerable fraction of national/regional emissions covered by both ODIAC-clump and X-clump (red bars). The sum of the fractions of TP (red bars) and TN (pink bars) are larger than 70% for all countries and regions, indicating that the algorithm applied to different maps allocates consistently the same groups of emitting grid cells into clumps. In Europe, the fraction of EDGAR emissions allocated to EDGAR-clumps (red plus blue bars in Fig. 8) is close to the fraction of ODIAC emissions allocated to ODIAC clumps (black line). In China, the fraction from MEIC is also close to that derived from ODIAC. But this fraction in PKU-CO$_2$-v2 (54%) is lower than that derived from ODIAC in China (84%). The differences between these fractions derived from ODIAC, MEIC and PKU-CO$_2$-v2 indicate large uncertainties in the distribution of emissions in China. This fraction in VULCAN (46%) is lower than that derived from ODIAC in USA (73%). In addition, in all regions, the fractions of emissions allocated to X-clumps (red plus blue bars) in X emission maps are all lower than those derived from ODIAC, indicating the emissions in ODIAC are more centralized toward populated areas than in other maps. This is attributed to the lack of line sources in ODIAC (Oda et al., 2018). The blue bars in Fig. 7, representing emissions from X maps that are not covered by ODIAC-clumps, are less than 10% of the total emissions in most cases, indicating that ODIAC-clumps miss only a small fractions of emission hotspots compared to other plausible fossil fuel CO$_2$ emission fields even without any adjustment. However, ODIAC-clumps would capture some low-emitting grid cells in other emission maps, as shown by the green bars in Fig. 8. Further investigation into the three types of clumps: ESRI clumps, non-ESRI clumps and point-sources clumps shows that the largest differences between ODIAC and X lie in the latter two types (Fig. S1-S3). The non-ESRI clumps account for a small fraction of the total emissions (less than 20% in general, Fig. 6 and S2), and the coherence in terms of fractions of emissions covered by non-ESRI clumps between different emission maps is less than 5% (red bars in Fig. S2). There are also large disagreements in the emissions from point-source clumps between different emission maps, as displayed by Fig. S3.

Figure 9 examines the consistency of the fractions of emissions covered by the same clumps between ODIAC and any emission map X. The consistency indicated by the red and pink bars is larger than 70%. The green bars are less than 10% in general, indicating that there are not many emission grid cells connecting different large cities. The major differences between ESRI clumps derived from various emission maps come from grid cells near the borders of ESRI clumps so that they are classified as ESRI clumps or other clumps in different emission maps (blue bars).





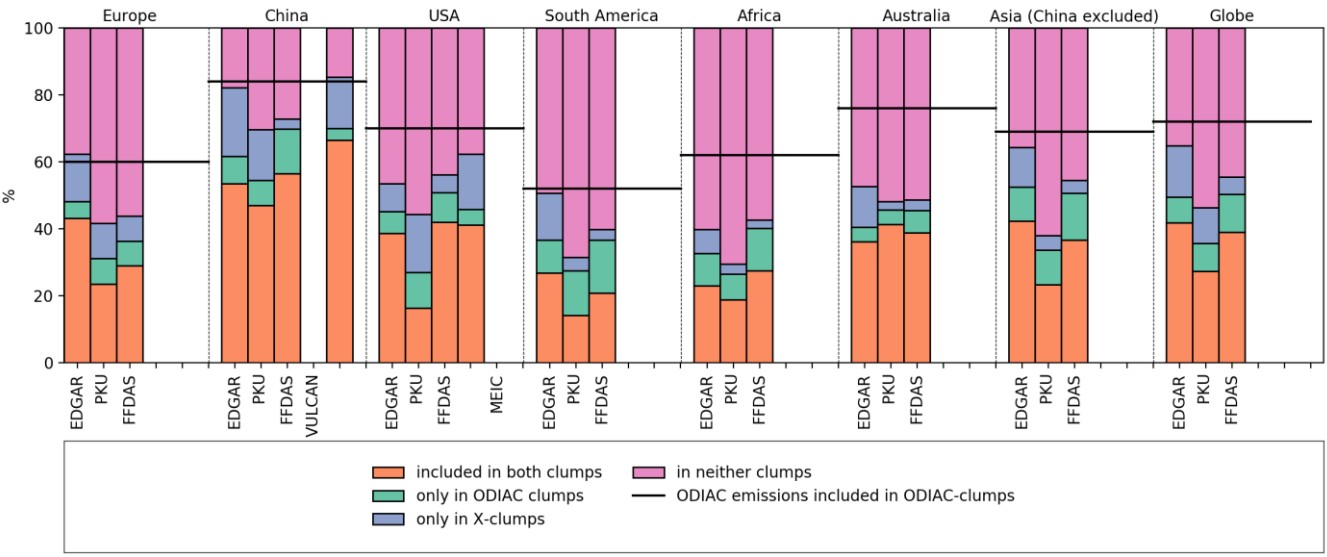

**Figure 8** The fractions of emissions from corresponding emission products covered: 1) by both ODIAC-clumps and X-clumps (red); 2) only by X-clumps but not by ODIAC-clumps (green); 3) by ODIAC-clumps but not by X-clumps (blue); and 4) by neither ODIAC-clumps nor X-clumps (pink). The thick black lines indicate the fractions of emissions in ODIAC covered by ODIAC-clumps.

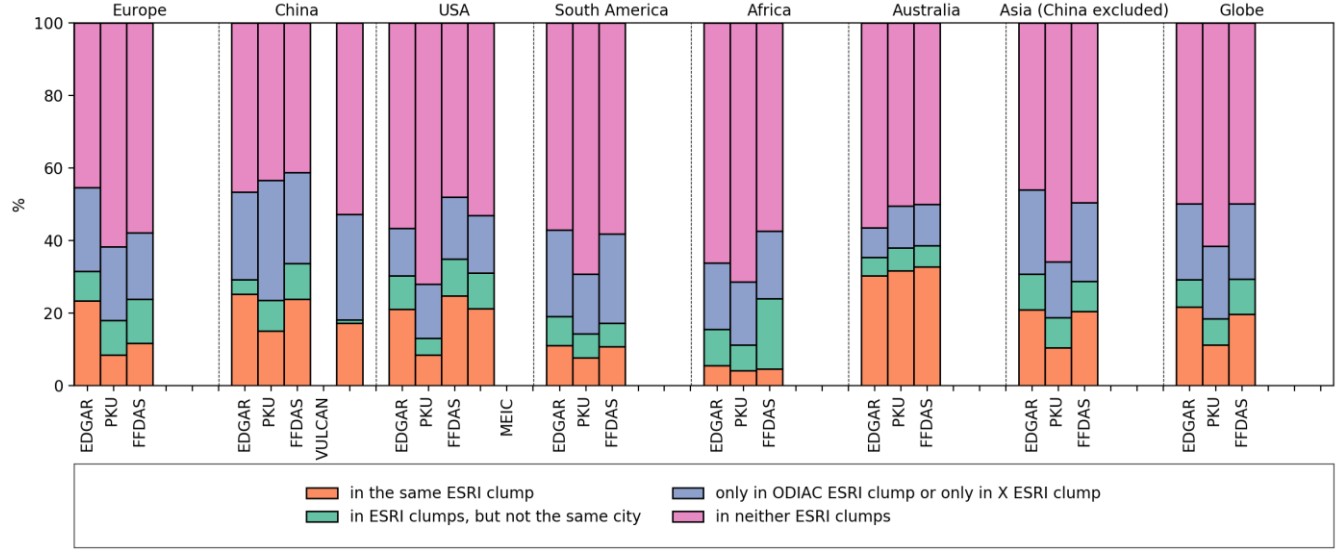

**Figure 9** The fractions of emissions from corresponding emission products covered 1) by the same ESRI clump from ODIAC and X (red); 2) by ESRI clumps in both ODIAC and X, but do not belong to the same ESRI urban area (green); 3) only by one of the ESRI clump in either ODIAC or X (blue); and 4) neither by any ESRI clump in ODIAC nor in X (pink).






## 4. Discussion

In this study, we use the map of urban area from ESRI and two thresholds to derive emission clumps. Threshold-1 determines the cores of the clumps, corresponding to a $XCO_2$ enhancement larger than the precision (0.5 ppm) of individual soundings without atmospheric horizontal transport (see Sect. 2.2 and Appendix). The precision largely depends on the

designs and configurations of different satellites. In this section, we test the sensitivity of the clumps to different assumptions on threshold-1 related to the precision of an individual sounding. The results listed in Table 3 show that the number of clumps are very sensitive to threshold-1, or individual $XCO_2$ sounding precision. However, the fractions of emissions covered by the clumps do not change significantly with threshold-1. The total number of clumps is reduced by 34% when the precision of an individual $XCO_2$ measurement is degraded to 1.0 ppm, compared to that obtained assuming 0.5 ppm, but the

fraction of emissions covered by all clumps is only reduced from 72% to 61%, e.g. 15% relative change. This indicates that a larger value of threshold-1 mainly removes clumps with small emissions. On the other hand, the number and fraction of emissions covered by point-source clumps are not sensitive to threshold-1, due to the fact that their emissions are highly concentrated in limited area. On the contrary, the number and emissions associated with non-ESRI clumps are the most sensitive to the precision.

Threshold-2 is used to define which grid cells shall be aggregated with the cores to form a clump. In this study, threshold-2 is chosen an order of magnitude smaller than threshold-1. This choice is somewhat arbitrary to include some marginal areas. With this default choice of threshold-2, the fraction of emissions from clumps to the total emissions is occasionally close to the estimate of the share of $CO_2$ emissions or energy use from cities to regional total in EIA and GEA (Table 1). The last two columns in Table 3 list the results for different values of threshold-2. Threshold-2 mainly impacts the

extent of surrounding grid cells near the cores of each area clump. When threshold-2 is chosen to be 0.071 g C m$^{-2}$ hr$^{-1}$ (twice as large as the default one), keeping threshold-1 as 0.36 g C m$^{-2}$ hr$^{-1}$, the fraction of emissions covered by the clumps to the global total is reduced from 72% (default result, T2) to 66%. The comparison between the results of T2, T6, and T4 in Table 3 shows that the identification of non-ESRI clumps is more sensitive to threshold-1 (precision), while the identification of ESRI clumps is more sensitive to threshold-2 (grid cells around cores in ESRI urban areas).





**Table 3** The sensitivity of number of emission clumps (integers before parentheses) and the fractions of emissions covered by the emission clumps (values in the parentheses) to global total to the thresholds in the clump algorithm

| Experiments | T1 | T2 | T3 | T4 | T5 | T6 |
|---|---|---|---|---|---|---|
| Precision of a single sounding (ppm) | 0.3 ppm | 0.5 ppm | 0.7 ppm | 1.0 ppm | 0.5 ppm | 0.5 ppm |
| Threshold-1 (g C m$^{-2}$ hr$^{-1}$) | 0.21 | 0.36 | 0.5 | 0.71 | 0.36 | 0.36 |
| Threshold-2 (g C m$^{-2}$ hr$^{-1}$) | 0.021 | 0.036 | 0.05 | 0.071 | 0.05 | 0.071 |
| ESRI clumps | 2756 (36%) | 2017 (32%) | 1498 (29%) | 1009 (26%) | 2017 (30%) | 2017 (28%) |
| Non-ESRI clumps | 6332 (15%) | 3071 (10%) | 1837 (7.7%) | 1109 (6%) | 3071 (9.2%) | 3071 (8.4%) |
| Point-source clumps | 6928 (30%) | 6226 (30%) | 5774 (30%) | 5304 (30%) | 6226 (30%) | 6226 (30%) |
| Total | 16016 (80%) | 11314 (72%) | 9109 (67%) | 7422 (61%) | 11314 (69%) | 11314 (66%) |

The emission clumps is a valuable concept relevant for the monitoring of fossil fuel $CO_2$ emissions from satellites. Fig. 8 shows that if the ODIAC-clumps are applied to other emission maps even without any adjustment, a majority of emission hotspots (indicated by red plus green bars in Fig. 8) are still included in the clump areas. However, Fig. 9 shows that there are large differences in the way emitting grid cells are grouped depending on the input emission map. When multiplying the map of ODIAC-clumps by another X emission map, the difference between the emissions from ODIAC and the emissions

from the same area in the X map, for a single clump, range between 0%-165% (5th - 95th percentiles). The relative differences tend to be larger for small clumps than large ones. For the monitoring of fossil fuel $CO_2$ emissions from the space, these results highlight the necessity to objectively associate the observed $CO_2$ plumes with underlying emitting regions.

The emission clumps defined in this study have at least one grid cell that will generate an excess of $XCO_2$ of at least 0.5 ppm over a morning period of 6 hours, assuming no atmospheric horizontal transport. This assumption is optimistic in terms

of detectability of $XCO_2$ plumes. In reality, accounting for wind advection or vegetation fluxes near a clump, $XCO_2$ enhancement in plumes may be smaller than 0.5 ppm, and therefore harder to detect with imagers. In this sense, the emissions covered in emission clumps derived based on such an assumption conservatively define the upper fraction of fossil fuel $CO_2$ emissions that could be constrained by $XCO_2$ imagery. In addition, the sampling of plumes will be reduced in presence of clouds, and will suffer from $XCO_2$ biases related to aerosol loads (Broquet et al., 2018; Pillai et al., 2016). The

emission clumps defined in this study provide a test bed for assessing the potential of satellite imagery for monitoring fossil fuel $CO_2$ emissions. In the future, global/regional inversion systems and observing system simulation experiment (OSSE) frameworks shall be developed using emission fields classified into clumps. Such inversions and OSSE studies will play a critical role in the deployment of new observation strategies and assessing the potential of these observing systems for assessing the fossil fuel $CO_2$ emissions (e.g. Broquet et al., 2018; Turner et al., 2016; Pillai et al., 2016).



**5. Data availability**

The ODIAC2017 data product is available from a website can be downloaded from the website http://db.cger.nies.go.jp/dataset/ODIAC/ (or https://doi.org/10.17595/20170411.001). The TIMES data product can be downloaded from http://cdiac.ess-dive.lbl.gov/ftp/Nassar_Emissions_Scale_Factors/. The clump map can be downloaded from https://doi.org/10.6084/m9.figshare.7217726.v1.

**6. Summary and Conclusion**

In this study, we have identified a set of large emission clumps from a high-resolution emission inventory. These clumps will generate individual atmospheric $XCO_2$ plumes that may be observed from space. This identification method identify the clump cores using ESRI map of major urban area and a high threshold related to the precision of $XCO_2$ measurements from planned satellites. It uses a low threshold and a RW algorithm to consider the area in the vicinity of the cores and split the area between different clumps based on the spatial gradients in the emission field. The emission clumps defined in this study depict the emitting hotspots around the globe that are relevant for the monitoring of fossil fuel $CO_2$ emissions from the satellites measurements. The clumps are derived with a trans-boundary approach, bypassing any artificial border imposed by national emissions accounting. In total, the emission clumps cover 72% of the total emissions in the original ODIAC. They defines the scales and regions of monitoring the short-term temporal profiles and long term trends in fossil fuel $CO_2$ emissions, which might be very useful for the Global Stocktaking exercise of UNFCCC. The clumps that have been identified here span a large range of emission. Given actual atmospheric transport, it is not clear whether those in the low range of emission generate an atmospheric $CO_2$ plume that can be identified from space. The presence of cloud cover may also challenge the detection of $XCO_2$ plumes and thus the estimate of emissions using space-borne measurements. Which fraction of the identified clump can be observed from space, and what accuracy can be expected from the atmospheric inversion requires an OSSE framework which shall be developed in a future paper.

**Appendix**

We make a calculation of the emission flux that would generate a 0.5 ppm excess of $XCO_2$ during 6 hours without wind. This is a conservative case with the accumulation of all emissions in the air column. The 0.5 ppm $XCO_2$ is taken as the individual sounding precision of a satellite $CO_2$ imager. Assuming a constant emission rate F (g C m$^{-2}$ hr$^{-1}$) during 6 hours, the $XCO_2$ excess ($XCO_2$, unit: ppm) is given by:

$$XCO_2 = F \times 6 / M_C / X_{air} \times 10^6 \qquad (1)$$

where $M_C$ ($= 12 \times 10^{-3}$ kg mol$^{-1}$) represented the molar mass of C, $X_{air}$ (unit: mol m$^{-2}$) represented the molar quantity of air mass in the air column. The $X_{air}$ could be approximated by:



$X_{air}=Psruf/g/M_{air}$ (2)

where Psurf ($=1.013\times10^5$ Pa) represents the surface pressure, g ($=9.8$ m s$^{-2}$) represents the acceleration of gravity, $M_{air}$ ($=29\times10^{-3}$ kg mol$^{-1}$) represents the average molar mass of air. Thus, the minimum emissions F* that would generate a 0.5 ppm excess of $XCO_2$ is computed: F*$=0.36$ g m$^{-2}$ hr$^{-1}$.

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
