# Peer review of "A global map of emission clumps for future monitoring of fossil fuel CO2 emissions from space"

_Earth System Science Data, 2018_

## Referee Comment (RC1) · Wang (Referee) · 3 Dec 2018

This study aims at delineating areas that can generate detectable atmospheric satellite CO2 plumes. A large fraction of fossil fuel CO2 emissions is coming from a very small fraction of the land surface, namely the emission clumps. Identifying these clumps is important for comparing fossil fuel CO2 emissions across different emission inventories and useful for observations of CO2 plumes from the space. To identify the clumps, the authors developed a random walker algorithm to consider the area in the vicinity of the cores and split the area between different clumps based on the spatial gradients in the emission field. In addition, a state-of-the-art CO2 emission inventory (ODIAC 2017) at a 30 x 30 arc-seconds is employed to perform this study.

This question is interesting. However, there is a question which is not answered by the authors. As pointed out by the authors, the emission clumps could be a useful tool to "provide a global dataset of fossil fuel $CO_2$ emission clumps for high-resolution atmospheric inversions that will use XCO2 imager data". However, these emission clumps are not fixed over time. The authors should explain how their emission clumps can be used in identifying hopspots in the future global satellite imagery of XCO2, when the emission clumps are evolving themselves from month to month and from year to year.

In addition, there are some necessary information that are missed in the paper. Please see some comments below. The paper can be accepted for publication after addressing all these comments.

Specific comments:

Line 21: For "cities and power plants", the relationship between the two concepts is ambiguous.

Line 23: Delete "very". I do not understand "closely". Rephrase "there is no detailed emission inventory for most of them". The English is not clear at some places, may be polished by a native speaker.

Line 27: Define "XCO2" before it is formally used.

Line 30: This sentence is too long, and should shorten for clarity.

Line 49: For "cities and power plants", the relationship between the two concepts is ambiguous. They may overlap.

Line 51: This sentence is too long, and should shorten for clarity.

Line 82: "a high resolution global map of fossil fuel CO2", it seems that the authors realized the importance of a high spatial resolution. However, what about the high temporal resolution? Does this affect the identification of emission clumps.

Line 131: Provide evidence that the emission spatial distributions do not change significantly from year to year.

Line 139-163: These two paragraphs are given under a section namely "ODIAC fossil fuel CO2 emission map". However, the authors are not introducing the ODIAC product here, and it confuses me a lot when I was reading these paragraphs. I cannot understand the relationship between the ODIAC product and LEO, OCO-2, GOSAT and GeoCARB imagers.

Figure 1: I do not quite understand this graph. How do the authors divide the total cumulative share of global annual emissions into power plants and area sources?

Line 176-216: The presentation of the method used to identify the emission clumps is not clear. I tried to follow the steps in these paragraphs, but I was stopped by "Firstly" (line 177), "Secondly" (line 184), "1)" (line 191), "2)" (line 196), "3)" (line 204), "4)" (line 211). The structure is unclear. Since the core of this paper is to present a new method, the authors need to convince us that this method could be easily performed and possibly repeated by other researchers.

Line 176-216: Meanwhile, the authors should publish the original code that is used to produce the emission clumps.

Line 196: What is the relationship between the emission clumps identified by step "1)", "2)" and "3)"? Saying that they are independent, I would re-name them as Category A, B, C, … or those names used in Figure 6.

Line 204: What is the difference between "cities" and "towns"?

Line 266: "The clump with largest emission is Shanghai" Is it total emission or emission density?

Line 272-274: I cannot understand these sentences. Shorten for clarity.

Line 280: There are other data sets for power plants in China better than the CARMA data set, e.g. the MEIC inventory, which has been noticed by the authors.

Figure 5: I do not understand why most emission clumps look like circles or dots in the four zoomed region. Is it a visual illusion?

Section 4: there are too many materials dropped in this part, among which the relationship is not very clear. If these discussions are independent, they can be organized in different parts under different headlines.

Line 466: "a set of large emission clumps" for a large area or a large emission?

Line 467: "This identification method identify" -> "This method identifies".

Line 476: "Given actual atmospheric transport," -> "Given actual atmospheric transport condition,".

---

## Referee Comment (RC2) · Peter Rayner (Referee) · 24 Feb 2019

This paper presents an algorithm for generating distributions of CO2 emission hot-spots based on a high-resolution proxy. It applies this algorithm to generate such a distribution for 2016. It assesses the sensitivity of the distribution to parameters in the algorithm.

The paper is probably in scope for ESSD. My only concern is that it adds value to an existing data product rather than generating significant new data itself. Its main contribution is likely to be the clumping algorithm it uses and I urge the authors to make the algorithm as well as the data available. The paper is also clearly written and presented.

[Figure]

I believe the paper makes a significant contribution. My main concern is some un-examined assumptions. Most crucially the underlying data set is not a true map of emissions but of emission proxies, mainly nighttime lights plus off-line estimates of emissions from power-stations. The spatial distribution of the proxy might well differ systematically from that of real emissions. In particular, there is a good chance that on-road emissions have greater spatial extent around emission cores than nighttime lights and may serve to amalgamate proximal clumps. This is testable now since the recent VULCAN product is available at the same resolution and includes these emissions. I recommend running the algorithm over VULCAN and ODIAC within the contiguous U.S. for the same year and comparing results.

Some specific comments

L140 I did not think the DMSP lights were available for 2016 but that ODIAC had switched to VIIRS.

L240 Probably there is no need to mention the python version though pointing out the package used is good. Note my firm suggestion above that the algorithm be made available.

---

## Author Comment (AC1) · 24 Apr 2019

**Response to comments on "A global map of emission clumps for future monitoring of fossil fuel CO₂ emissions from space" by Y. Wang et al.**

We thank the referee for reviewing our manuscript. Please find attached a point-by point reply (in black) to each of the comments raised by the referee (in blue) with legible text and figures organized along the text. For your convenience, changes in the revised manuscript are highlighted with dark red. All the pages and line numbers correspond to the original version of text.

This study aims at delineating areas that can generate detectable atmospheric satellite CO2 plumes. A large fraction of fossil fuel CO2 emissions is coming from a very small fraction of the land surface, namely the emission clumps. Identifying these clumps is important for comparing fossil fuel CO2 emissions across different emission inventories and useful for observations of CO2 plumes from the space. To identify the clumps, the authors developed a random walker algorithm to consider the area in the vicinity of the cores and split the area between different clumps based on the spatial gradients in the emission field. In addition, a state-of-the-art CO2 emission inventory (ODIAC 2017) at a 30 x 30 arc-seconds is employed to perform this study. This question is interesting. However, there is a question which is not answered by the authors. As pointed out by the authors, the emission clumps could be a useful tool to "provide a global dataset of fossil fuel CO2 emission clumps for high-resolution atmospheric inversions that will use XCO2 imager data". However, these emission clumps are not fixed over time. The authors should explain how their emission clumps can be used in identifying hotspots in the future global satellite imagery of XCO2, when the emission clumps are evolving themselves from month to month and from year to year.
In addition, there are some necessary information that are missed in the paper. Please see some comments below. The paper can be accepted for publication after addressing all these comments.

**Response:**

We would like to thank the referee for the valuable comments and suggestions for improving our manuscript.

The specific concern raised by the reviewer about the temporal variation in the emission spatial distribution and thus the identification of clumps are discussed in the reply to Comment 7 and 8. The clump map derived in this paper is based on ODIAC for the year 2016, and it could be updated annually once new versions, either from ODIAC or other high-resolution emission inventories, become available. The purpose of the study was to propose and test a new algorithm to determine clumps and to provide the global clump map for a typical year, not to analyze trends in clumps, which could be a topic for a follow up study.

Specific comments:
1) Line 21: For "cities and power plants", the relationship between the two concepts is ambiguous.

**Response:**

"City" emission refers to emission other than those from power plants. Oda et al. (2018) separated the emissions from power plants (called point sources in ODIAC) and other

emissions (called non-point sources in ODIAC). The non-point sources were distributed in proportion to the nighttime light that is observed from space. To clarify the relationship between "cities" and "power plants", we revise the sentence into: "A large fraction of fossil fuel $CO_2$ emissions occur within "hotspots", such as cities (where direct $CO_2$ emissions related to fossil fuel combustion in transport, residential, commercial sectors, etc., excluding emissions from electricity-producing power plants, occur), isolated power plants, and manufacturing facilities, which cover a very small fraction of the land surface…."

2) Line 23: Delete "very". I do not understand "closely". Rephrase "there is no detailed emission inventory for most of them". The English is not clear at some places, may be polished by a native speaker.
**Response:**
We delete "very".

We revised this sentence: "…small fraction of the land surface. The coverage of all high-emitting cities and point sources across the globe by bottom-up inventories is far from complete, and for most of those covered, the uncertainties in $CO_2$ emission estimates in bottom-up inventories are too large to allow continuous and rigorous assessment of emission changes (Gurney et al., 2019). Spaceborne imagery of…"

We have requested several fluent English speakers to take more active roles in proofreading the whole manuscript. We hope the revised manuscript could satisfy your concerns.

3) Line 27: Define "XCO2" before it is formally used.
**Response:**
We revise the sentence: "The proposed space-borne imagers with global coverage planned for the coming decade have a pixel size on the order of a few square kilometers, and an accuracy and precision of <1 ppm for individual measurements of vertically integrated columns of dry air mole fractions of $CO_2$ ($XCO_2$)."

4) Line 30: This sentence is too long, and should shorten for clarity.
**Response:**
We revise the sentence: "In this study, we  characterize area and point fossil fuel $CO_2$ emitting sources  generating coherent $XCO_2$ plumes  that may be observed from space . We characterize these emitting sources around the globe and they are referred to as "emission clumps" hereafter. An algorithm is proposed…"

5) Line 49: For "cities and power plants", the relationship between the two concepts is ambiguous. They may overlap.
**Response:**
We revise the sentence: "The contribution from cities (excluding electricity-related emissions from large power plants, see Sect. 2) and power plants to national and global mitigation efforts is thus critical (Creutzig et al., 2015; Shan et al., 2018)."

**Response:**

We revise the sentence: "The technique called atmospheric $CO_2$ inversion quantifies emissions based on a prior estimate from inventories, atmospheric $CO_2$ measurements and atmospheric transport models. Inversions of fossil fuel CO2 emissions have…"

7) Line 82: "a high resolution global map of fossil fuel CO2", it seems that the authors realized the importance of a high spatial resolution. However, what about the high temporal resolution? Does this affect the identification of emission clumps?

**Response:**

We are aware of that there are some day-to-day and month-to-month variations in the spatial distribution of fossil fuel $CO_2$ emissions. And these variations can lead to variations in the spatial extent of the clump over the same timescales. In this study, we arbitrarily choose threshold-2 to be an order of magnitude smaller than threshold-1 to include marginal areas. We assume that our conservative definition of threshold-2 ensures that the effective clumps (clump cores) always stay between our boundaries, and it also ensures the consistency of the unit for which the satellite observations could provide emission estimates through one year.

In addition, in the regions experiencing fast urbanization, we agree that the clump definition should be updated annually based on the latest inventories at high resolution to track the trends in growing cities.

To address the reviewer's concern, we revise the manuscript by:

- Adding in Line 162: "…For calculating clumps based on morning emissions, we multiplied the annual mean emission rate (unit: g C m$^{-2}$ hr$^{-1}$) in each grid cell of ODIAC by the average scaling factors of emissions between 6:00-12:00 local time. The day-to-day and month-to-month variations in the spatial distribution of fossil fuel $CO_2$ emissions may lead to temporal variations in the spatial extent of the clumps. In this study, we define the clumps based on two thresholds (see Sect. 2.2) to ensure that the effective clumps are always within the boundaries of the clumps, and that the satellite observation should provide emission estimates consistently within a year. We thus ignore the month-to-month and day-to-day variations in the emissions."

- Adding in L422: "… In this study, threshold-2 is chosen an order of magnitude smaller than threshold-1. This choice is somewhat arbitrary to include some marginal areas. Such marginal area accounts for the fact that the outskirt of the cities could also contribute to the city cores. In addition, the marginal area ensures that the effective clumps (e.g. the cores of the clumps) will always be accounted for in the clump map within a short time span (typically within one year to among few years). With this default choice of threshold-2, …".

- Adding in Line 447: "…, these results highlight the necessity to objectively associate the observed $CO_2$ plumes with underlying emitting regions. In this study, the clumps are only defined based on the ODIAC emission map for the year 2016. However, in the regions experiencing fast urbanization rates, the spatial distribution of emissions are also changing rapidly. In order to build an operational observing system in the near future, it is also necessary to consistently update the clump definition based to latest

emission maps to track the trends in the emissions and $CO_2$ plumes for growing cities."

8) Line 131: Provide evidence that the emission spatial distributions do not change significantly from year to year.

**Response:**

We are aware that some significant spatial distribution changes may be expected for countries with fast urbanization, e.g. China and East Asia (Frolking et al., 2013). As discussed in the response to Q7, our algorithm include some marginal areas around the city cores, ensuring that the clump map always capture the effective clumps in a short period (within one year to few years). In addition, the algorithm can be applied for each year to track the year-to-year variations in the clump definition for growing cities.

We add in L131: "… We chose the year 2016 assuming that the emission spatial distributions do not change significantly from year to year. In regions with rapid urbanization rates, the emission spatial distributions may change rapidly. The analysis of such changes is out of the scope of this paper, but the clump definition can be updated consistently with the latest high-resolution emission maps for each year, using the approach presented in Sect. 2.2. The ODIAC dataset provides…"

9) Line 139-163: These two paragraphs are given under a section namely "ODIAC fossil fuel CO2 emission map". However, the authors are not introducing the ODIAC product here, and it confuses me a lot when I was reading these paragraphs. I cannot understand the relationship between the ODIAC product and LEO, OCO-2, GOSAT and GeoCARB imagers.

**Response:**

The ODIAC map provides the emission field at a high spatial resolution. However, ODIAC has a monthly temporal resolution. Because the planned LEO satellite imagers fly over a city at a local time close to noon, the plumes that impact the $XCO_2$ imagery are generated by morning emissions. To correctly link the $XCO_2$ observations and underlying emissions, morning emissions rather than the monthly mean emissions are needed. In this case, we apply the hourly profiles from TIMES to estimate the morning emissions from the monthly means. The overpass time of considered satellites determines how the ODIAC product is used.

To remove the ambiguity identified by the reviewer, we revise the manuscript by:

- Changing the title of section to "2.1 Development of a high resolution emission map of morning emissions".

- Moving the paragraph L139-L147 into the introduction L116: "Because $CO_2$ produced by emissions is quickly dispersed by transport, $XCO_2$ plumes sampled at a given time by a satellite image usually relate to emissions that occurred few hours before its acquisition (Broquet et al., 2018). In this study, we focus on planned LEO imagers on Sentinel missions, assuming an equator crossing time around 11:30 local time (Buchwitz et al., 2013; Broquet et al., 2018) so that $XCO_2$ plumes sampled by these imagers are from morning emissions. Different overpass times are also possible for other satellites. For example, Equator crossing times of OCO-2 and GOSAT are 13:00-13:30 local time. Geostationary imagers may provide a better temporal coverage of the emissions; e.g. GeoCARB images are considered to sample a city for multiple times within a day

(O'Brien et al., 2016)."

- Deleting Line 146: "  To estimate morning emissions, …"

10) Figure 1: I do not quite understand this graph. How do the authors divide the total cumulative share of global annual emissions into power plants and area sources?

**Response:**

ODIAC emissions can be provided in several emission types, such as point source (power plant) and non-point sources. The locations and emissions for power plants in ODIAC are derived from the CARDA dataset, while the area emissions are computed by distributing the national emissions other than power plants based on the nighttime light.

Fig. 1 is plotted based on such separation of emissions from power plants and area sources. The cumulative share of the area emissions are shown in blue shade, and the cumulative share of the emissions from power plants are plotted in red shade and on top of the share of area emissions.

We add in line 165: "In ODIAC, the point sources only refer to power plants in the CARMA database. So in this study, we refer to sources other than power plants as area sources. Before clumps are calculated, …"

11) Line 176-216: The presentation of the method used to identify the emission clumps is not clear. I tried to follow the steps in these paragraphs, but I was stopped by "Firstly"(line177), "Secondly"(line184),"1)"(line191),"2)"(line196),"3)"(line204),"4)"(line211).The structure is unclear. Since the core of this paper is to present a new method, the authors need to convince us that this method could be easily performed and possibly repeated by other researchers.

**Response:**

Figures 2 and 3 may be used for a further understanding of the algorithm. "Firstly, ..." corresponds to the selection of the power plants. "Secondly, …" corresponds to the computation of area clumps. The computation of area clumps follows four steps numbered 1) to 4).

To make it clearer, we add in line 176 "… Fig. 3 illustrates how it operates for a small domain as an example. Two categories of emission clumps are defined. A)  … B) …The four steps to compute area sources emission clumps are detailed as below." In addition, we provide the code with detailed comments for the algorithm.

12) Line 176-216: Meanwhile, the authors should publish the original code that is used to produce the emission clumps.

**Response:**

We do provide the code, with detailed comments, that is used to produce the emission clumps.

13) Line 196: What is the relationship between the emission clumps identified by step "1)", "2)"and "3)"? Saying that they are independent, I would re-name them as Category A, B, C, …or those names used in Figure 6.

**Response:**

    The emission clumps are categorized in three types. The first one is the power plants (point-source clumps) which was directly taken from the power plants in ODIAC (which was based on CARMA dataset) (step "A)" in the revised manuscript, see the response to Q11). The second and third types of clumps are both area clumps and they are defined in step "B)" and steps "1)"-"4)" (see the response to Q11 and the revised manuscript). The clumps in the second type all have a core which was identified as "urban" by ESRI urban map. The clumps in the third type do not have a core from ESRI urban map, but have a core whose emissions are above threshold-1. Step "1)" defines the potential grid cells to be included in any clump, but does not define any clump. Step "2)" defines the cores correspond to an ESRI urban areas. Step "3)" defines other cores outside the ESRI urban area. Step "4)" attributes the grid cells defined in step "1)" and outside the area in steps "2)" and "3)" to either ESRI cores or non-ESRI cores.

    We prefer to call them "point-source clump", "ESRI clump" and "non-ESRI clump", because they are more detailed than just "A", "B" and "C". Because the whole text discussed the differences and features of the three clumps, using these detailed names will help readers without any difficulty recalling what "A" and "B" and "C" refer to.

    We add in line 216: "…recognizing different segments/objects in a picture or photograph. The clumps with an ESRI core (step 2) are called "ESRI clumps", while the clumps with a non-ESRI core (step 3) are called "non-ESRI clumps" hereafter. This step is illustrated in Fig. 3e…

**14) Line 204: What is the difference between "cities" and "towns"?**

**Response:**

    We find that the ESRI urban map is not complete to cover all the emission hotspots. For example, Fig. 4b in the manuscript show that ESRI urban map only identify Beijing, Tianjin, Langfang, etc. But we do see some areas with strong emissions outside these ESRI-urban area from Fig. 4e in the manuscript. Because these areas are usually smaller (in terms of surface area) than the ESRI-urban area, and is not connected to ESRI-urban area, we call them "smaller populated area". Because countries differ in the levels of administrative divisions, to avoid the confusion raised by the referee, we changed the word "towns" into "small cities".

**15) Line 266: "The clump with largest emission is Shanghai" Is it total emission or emission density?**

**Response:**

    It is the total emission. We revise the sentence: "… The clump with largest annual emission budget is Shanghai, which emits 47 Mt C per year.…"

**16) Line 272-274: I cannot understand these sentences. Shorten for clarity.**

**Response:**

    We revise the sentence into: "… This is because the southeast coast of China is densely populated even within rural  areas (yellow-green outside the urban area of ESRI urban map in Fig. 4e), and because the emission rates per capita is also high in China compared to the world average (Janssens-Maenhout et al., 2017). As a result, our algorithm finds more

non-ESRI clumps and larger area for each clump in China than other regions."

**Response:**

MEIC inventory is used for comparison in Sect. 3.2. In this part, we only discussed the distribution of clumps based on the ODIAC map, because it is a global map that were derived consistently among countries. In the paper, we aim at some consistency between regions that allow comparisons. On the other hand, the methodology that is described in the paper can be applied to MEIC (China) or VULCAN (USA) datasets, among others, for more regional applications.

**Response:**

In this figure, we only plotted the location with small dots rather than the size and shape of the clumps. At continental scale, it is hard to display the precise shape of the clumps, which was shown by Fig. 4 in the manuscript. To make it clearer, we revise the title of this figure: "**Figure 5** The spatial distribution of emission-weighted center of the emission clumps all over the globe. The inserted plots zooms over 4 regions that contain most of the clumps."

**Response:**

Following the reviewer's suggestion, we rewrite this section with three sub-sections: "4.1 Impacts of the sounding precision on the identification of emission clumps" (with first two paragraphs in the original manuscript), "4.2 Impact of using ODIAC on the identification of emission clumps" (a newly added paragraph and the third paragraph in the original manuscript), "4.3 Implication for future inversion studies" (the last paragraph in the original manuscript).

**Response:**

We revise the sentence: "In this study, we have identified a set of  emission clumps with large emission rates (in the unit of g C m$^{-2}$ hr$^{-1}$) from a high-resolution emission inventory. ..."

**Response:**

We revise the sentence as the reviewer suggested.

22) Line 476: "Given actual atmospheric transport,"-> "Given actual atmospheric transport

condition,".

**Response:**

We revise the sentence as the reviewer suggested.

---

## Author Comment (AC2) · 24 Apr 2019

**Response to comments on "A global map of emission clumps for future monitoring of fossil fuel CO₂ emissions from space" by Y. Wang et al.**

We thank the referee for reviewing our manuscript. Please find attached a point-by point reply to each of the comments raised by the referee with legible text and figures organized along the text. Please find below the point-to-point responses (in black) to all referee comments (in blue). For your convenience, changes in the revised manuscript are highlighted with dark red. All the pages and line numbers correspond to the original version of text.

This paper presents an algorithm for generating distributions of CO2 emission hotspots based on a high-resolution proxy. It applies this algorithm to generate such a distribution for 2016. It assesses the sensitivity of the distribution to parameters in the algorithm.
The paper is probably in scope for ESSD. My only concern is that it adds value to an existing data product rather than generating significant new data itself. Its main contribution is likely to be the clumping algorithm it uses and I urge the authors to make the algorithm as well as the data available. The paper is also clearly written and presented.

**Response:**

We would like to thank the referee for the valuable comments and suggestions for improving our manuscript.

Indeed, the algorithm presented in this paper is one of the major assets of this paper. The algorithm can be applied to other high-resolution emission maps (Sect. 3.2). Apart from the algorithm itself, Sect. 3.2 showed some consistencies between the results derived from ODIAC and those based on other emission maps. Given such consistencies algorithm, the complexity and the value of the algorithm, we think that this paper is in scope for ESSD: "Articles on methods describe nontrivial statistical and other methods employed (e.g. to filter, normalize, or convert raw data to primary published data) as well as nontrivial instrumentation or operational methods." (https://www.earth-system-science-data.net/about/aims_and_scope.html).

We have published the dataset at https://doi.org/10.6084/m9.figshare.7217726.v1. And we provide the code in the supporting information.

I believe the paper makes a significant contribution. My main concern is some unexamined assumptions. Most crucially the underlying data set is not a true map of emissions but of emission proxies, mainly nighttime lights plus off-line estimates of emissions from power-stations. The spatial distribution of the proxy might well differ systematically from that of real emissions. In particular, there is a good chance that onroad emissions have greater spatial extent around emission cores than nighttime lights and may serve to amalgamate proximal clumps. This is testable now since the recent VULCAN product is available at the same resolution and includes these emissions. I recommend running the algorithm over VULCAN and ODIAC within the contiguous U.S. for the same year and comparing results.

**Response:**

We appreciate that the reviewer confirm the contribution of this study to the community.

The suggestion by the reviewer is indeed an important one. We are aware that we can't assume an emission field from a single emission dataset as perfect. As pointed out by the reviewer, ODIAC might miss some of the on-road emissions in the emission distribution due to the use of nightlight emission proxy. Following the reviewer's suggestion, we run the algorithm over the new version of the VULCAN emission data product (VULCANv3.0) provided by Prof. Gurney, one of the co-authors of this manuscript, and compare the results with the one based on ODIAC. The VULCANv3.0 use detailed primary data sets across the US. In the new version of VULCAN, they are in principal the same collection of datasets as described in Gurney et al. (2009), but with improvements in the data quality. The VULCANv3.0 also improves the spatial and temporal resolution compared to VULCANv2.2 (Gurney et al., 2009).

Fig. R1 shows the clump results based on ODIAC (a-f) and VULCANv3.0 (g-l) in the vicinity of three mega cities in US. The emission field in ODIAC are much smoother than that in VULCANv3.0. In VULCANv3.0, there are a large amount of small clumps around the large cities. Some of these small clumps correspond to the on-road emissions (e.g. long and narrow lines), and some correspond to small cities. For the on-road emissions, the algorithm sometimes split the road into several segments (e.g. the Pacific Coast Highway, Fig. R1h). In total, the ODIAC clumps covers 58% of the emissions in VULCANv3.0, while the emissions from on-road transportation and small cities that are missed by ODIAC clumps account for 27% of the total emissions in VULCANv3.0. This result is similar to that discussed in the manuscript Sect. 3.2, indicating some consistencies between the clump results derived from different emission products.

We would like to note that VULCANv3.0 is not yet publicly available, and that ESSD does not recommend to include such data (see Carlson and Oda, 2018 ESSD). Following the recommendation by the editor, we have not included this comparison in the manuscript. However, we discuss the limitation of the single use of the ODIAC product, which used nighttime light as a proxy for emissions in Sect. 4.2.

"**4.2 Impact of using ODIAC on the identification of emission clumps**

ODIAC used nighttime light as a proxy for the spatial distribution of emissions. The accuracy of the proxy in representing the distribution of actual emissions largely impacts the extent of the clumps. For example, compared with other emission products, ODIAC does not capture line source emissions such as on-road transportation (Oda et al., 2018; Gurney et al., 2019). The satellite observations of CO indicated significant CO enhancement over major roads (Borsdorff et al., 2019). Since our clump map is derived from ODIAC emission product, some of the roads that generate significant $XCO_2$ plumes may be missed by the clumps defined in this study. As the ODIAC team is planning to include transportation network data in their emission product (Oda et al., 2018), our clump map could be updated with a new version of ODIAC,

 Fig. 8 shows that… "

[Figure]

**Figure R1** Emission clumps near New York (a, d, g and j), Los Angles (b, e, h and k) and Chicago (c, f, i and l) based on ODIAC product (a-f) and VULCANv3.0 (g-l). In a-c and g-i, solid lines depict the urban areas from ESRI product. Colored patches depict the clump area. In d-f and j-l, solid lines depict the boundaries of final clumps (boundary of colored patches in a-c and g-i). Colored fields in d-f show the emissions from ODIAC product. Colored fields in j-l show the emissions from VULCANv3.0. Light dashed lines indicate 1 °×1 °grids.

Some specific comments
1) L140 I did not think the DMSP lights were available for 2016 but that ODIAC had switched to VIIRS.
**Response:**

The ODIAC model employs the DMSP radiance calibrated nighttime light products (https://www.ngdc.noaa.gov/eog/dmsp/download_radcal.html) for estimating emission spatial distributions of non-point emissions (see Oda et al. 2018). As the reviewer pointed out, the DMPS data are not available for the year 2016 (the latest radiance calibrated data is for year 2010).    The current ODIAC model uses the 2010 DMSP nighttime light product for the period 2010-2017. As mentioned in Oda et al. (2018), the research team plans to use the VIIRS nightlight for future versions of the ODIAC emission product development. But the version of the ODIAC data product used in this study (ODIAC2017) is still based on the DMSP nighttime light data.

2) L240 Probably there is no need to mention the python version though pointing out the package used is good. Note my firm suggestion above that the algorithm be made available.
**Response:**
To maintain the traceability and reproducibility, we provide all the computer codes that is used to produce the emission clumps presented in this study, with detailed comments.

**References:**

Carlson, D. and Oda, T.: Editorial: Data publication – ESSD goals, practices and recommendations, Earth System Science Data, 10(4), 2275–2278, doi:https://doi.org/10.5194/essd-10-2275-2018, 2018.

Gurney, K. R., Mendoza, D. L., Zhou, Y., Fischer, M. L., Miller, C. C., Geethakumar, S. and Can, S. de la R. du: High Resolution Fossil Fuel Combustion CO2 Emission Fluxes for the United States, Environmental Science & Technology, 43(14), 5535–5541, 2009.